# Identifying the Characteristics of Non-Digital Mathematical Games Most Valued by Educators

**James Russo** [1,*], **Leicha Bragg** [2], **Toby Russo** [3] **and Michael Minas** [4]

1. School of Curriculum Teaching & Inclusive Education, Monash University, Clayton 3800, Australia
2. Victoria University, Footscray 3011, Australia
3. Fitzroy North Primary School, Fitzroy North 3068, Australia
4. Love Maths, Williamstown 3016, Australia
* Correspondence: james.russo@monash.edu

**Abstract:** Non-digital games are frequently used to support primary mathematics instruction. Moreover, we know from the literature that to increase the likelihood that a chosen mathematical game is educationally rich it should reflect specific principles, such as offering a balance between skill and luck and ensuring that a key mathematical focus is central to gameplay. However, there is limited research informing us, from a teacher's perspective, of the specific characteristics of mathematical games that are most indicative of a game's value for supporting learning, and the likelihood that teachers will use the game with students in the future. To help address this gap, the current study invited 122 educators to complete an on-line questionnaire, including 20 Likert-scale items designed to assess the characteristics of educationally-rich mathematical games (CERMaGs) that aligned with six 'good practice' principles previously identified in the literature, in relation to a specific mathematical game of their choosing. In total, educators chose a broad range of mathematical games to be evaluated (n = 53). On average, they reported that their chosen game was highly valuable for supporting mathematics learning and that they were very likely to use this game with students if given the opportunity. Our results revealed that the extent to which educators perceived a game to be suitably challenging, engaging, enjoyable, modifiable to support different learners, and transformable into an investigation or broader mathematical inquiry, were particularly important characteristics associated with perceptions of a game's educational value. Similarly, perceived levels of student enjoyment, engagement and a game's potential to lead to a rich mathematical investigation were important characteristics for evaluating the likelihood that an educator would use a particular game in the future with students if given the opportunity, as was the capacity of a game to support mathematical discussion. The implications of these findings for supporting classroom practice and teacher professional learning are discussed.

**Keywords:** mathematics education; games; student engagement; teacher perspectives

## 1. Introduction

Gameplaying, as a social endeavour, is an integral part of all cultures. The oldest known board game, the Royal Game of Ur, was played 4600 years ago in ancient Mesopotamia [1]. Gameplaying has endured over thousands of years. Recently, due to the global pandemic, there has been a global resurgence of people engaging in both digital and non-digital games. Euromonitor International's market analysis estimated a boom in the games and puzzles market of almost AUD 1 billion in value in the first year of the global pandemic, compared to the previous year [2]. The market for both digital and non-digital games continues to climb, despite the perception that digital games would eliminate or severely detract from non-digital game usage. The persistence of non-digital games may be a product of their relatively lower cost, as well as the difficulties accessing the necessary technology. However, Fang et al.'s [3] study comparing the emotional reactions of players



between traditional (non-digital) board games and their digital counterparts, found that positive emotions are more strongly aligned with traditional board games. The affection towards non-digital games likely also explains, in part, their endurance. In our recent study, we found that more than three-quarters of Australian primary school teachers used games multiple times per week to support mathematics instruction, and that these teachers were far more likely to prefer employing non-digital games in the classroom, compared with digital games [4]. Other research has revealed that teachers view non-digital mathematical games as a means of enhancing engagement with the broader school community, and involving parents in their child's mathematics education through initiatives such as mathematics games days [5,6]. The teachers' valuing of, and preferences for utilising, specific types of mathematical learning activities is a particularly relevant consideration in countries such as Australia, where the curriculum is not prescriptive and teachers have a great deal of autonomy to develop a specific learning program that meets their students' needs [7,8]. The continued engagement and enjoyment of non-digital games led to us questioning which characteristics of educationally rich mathematical games are most valued by educators. This is the focus of the current study.

Our earlier review of the literature identified five principles of educationally rich mathematical games [9]; students are engaged, skill and luck, mathematics is central, flexibility for learning and teaching, and, home-school connections. A sixth principle, games into investigations, was subsequently identified [10]. These six principles of educationally rich mathematical games are described below.

### 1.1. Students Are Engaged

Engagement is predictive of students' achievement in mathematics, with a lack of engagement potentially impacting students' performance [11]. Australian students are increasingly demonstrating disengagement with mathematics and a decline in mathematical aspirations, as they move from primary education through to high school [12]. Thus, engagement is an advantageous characteristic of learning tasks.

Gameplay is typically viewed as being an enjoyable activity when players become captivated with the experience. Similar to the use of serious (digital) games to leverage entertainment qualities for training and educational purposes [13,14], educators capitalise on the pleasurable characteristics of games to engage their students in mathematical content. Games offer an avenue for flow, " . . . a state of concentration so focused that it amounts to the absolute absorption in an activity." [15] (p. 1). A key dimension of flow is achieving a balance between the challenge of the activity and the individual's skills [16]. This balance leads us to the characteristic of skill and luck.

### 1.2. Skill and Luck

A key ingredient in defining a game is its indeterminacy; not knowing the outcome is a common feature of games [17]. Playing a game where the outcome is known prior to commencing removes the challenge that many players seek. Being optimally challenged enhances engagement, whereas a lack of challenge may result in boredom and disengagement [18,19]. Randomness in a game is what determines the uncertainty of the outcome. The use of dice, a spinner, or drawing a card, are tools to facilitate randomness. Games of chance are employed in the mathematics classroom to explore all mathematical concepts, and strongly align with probability and statistics. The foundations of probability were first understood as a result of mathematician and gambler Gerolamo Cardano's investigation of dice games, around 1564 [20], and further developed and formalised by mathematicians Blaise Pascal and Pierre de Fermat a century later [21]. Games of chance are the backbone of the gambling industry with players of all skill levels having an equal chance of the same outcome, due to randomness [22]; whereas games of skill depend on mental dexterity [22], and in the case of classroom games, favour students who possess stronger mathematical capabilities than their opponents. There is evidence that games that rely solely on skill, particularly when combined with time pressure and an emphasis on speed, can reduce

student motivation to play and increase stress levels, when compared with games that also incorporate a luck element [23].

Games of chance and games of skill are not diametrically opposed to one another, but rather games fall along a continuum, from games that are only determined by luck, e.g., Snakes and Ladders, through to those games that are heavily skills based, e.g., Chess. The manipulation of games of luck to offer an element of player choice can introduce an aspect of skill [24]. Games of skill do incur an element of unpredictability, as they are dependent on human intervention, and humans are not predicable [17]. Skilled players can have a lapse in judgement. Finding a balance between luck and skill within a mathematical game is important in building the students' mathematical understanding, interest, and providing opportunities for all students to experience both winning and losing [9].

### 1.3. Mathematics Is Central

Mathematics may be overt in games or obscured and require unpacking by the teacher to ensure the mathematics is explicit; however, fundamentally, mathematics must be a central characteristic of a game for it to be considered a mathematical game. Employing games in the mathematics classroom has a primary objective of introducing, practicing and/or extending mathematical knowledge [9]. Games are often utilised to foster students' fluency to support mathematics proficiency and achievement [25,26], and extend students' reasoning and understanding through justifying strategies and solutions [27,28]. Games enhance students' confidence in their mathematical capabilities and excite them about mathematics [29]. Gameplay is an effective practice for students in need of intensive intervention in mathematics [30]. The selection of the game must be purposeful and considered to enhance students' mathematical learning.

### 1.4. Flexibility for Learning and Teaching

Have you played a well-known game at a friend's home, only to discover their game rules differ to the official rules? They claim "house rules", where unofficial rules are adapted and adopted by those playing. House rules is an example of differentiating the game to suit a group's needs and challenge level. Games played inside and outside the classroom have a common feature in that they have the capacity to be adapted and altered by the players, and importantly, all players are aware of the revised rules. The adaptability of a game's mechanics, meaning the rules and materials, offers the potential to differentiate for the mathematical needs of the players [10], and are one of the reasons that games, as a mathematics learning tool, can be used with such diverse groups of students, including students with intellectual disabilities [31] and mathematically gifted students [32]. Game mechanics provide a flexibility that can increase and decreases the level of challenge. Similar to the call for board game designers to increase the level of difficulty and strategic thinking required in games to foster challenge and flow [33], teachers too should consider how they can modify games to promote deep mathematical thinking and optimise challenge [9].

### 1.5. Home-School Connections

A child's first teacher is their family, whose expectations and attitudes are influential in their child's mathematical learning [34]. A meta-analysis of 64 quantitative studies revealed that the home mathematics environment, particularly parent and child mathematical interactions, are positively associated with mathematics achievement [35]. Berkowitz et al.'s [36] study of 587 first grade families engaging in an intervention of reading short numerical story problems at home, showed a significant increase in children's mathematics achievement over the school year. A recent study [37] of 50 pre-school children and their families playing card games within their home environment revealed a significant positive change in the recognition and matching of shapes. Conversations between family members and children about mathematical concepts during gameplay facilitated children's mathematical learning [38]. Not surprisingly, in the same study, it was found that the longer duration and increased frequency of gameplaying impacted positively on the children's

learning. Importantly, engaging families in mathematics at home builds connections between school and home, thereby fostering a link between the applicability of classroom learnt mathematics to home and community mathematical experiences [39].

### 1.6. Games into Investigations

Games offer a unique opportunity for rich investigation of the mathematics underlying the mechanics and strategies of a game [10]. Exposure to investigating games of chance equips students with insights into misconceptions associated with probabilities and randomness, particularly related to gambling [40]. Based on a long history of promoting the use of games in the primary classroom, Swan and Marshall [41] designed a series of probability experiments that utilise games of chance to compare and contrast fair and unfair games. A key objective of educational strategies that investigate fair and unfair games and probability misconceptions, such as the gambler's fallacy, is to support the informed and responsible choices for those who may gamble in the future [42]. While games of chance are well-suited to probability investigations, games in general offer a diversity of mathematical concepts to examine across all ages. Byrne's [43] study of four under-graduate students engaged in an inquiry-based course of exploring the mathematics within commercial games, demonstrated an increase in participants' mathematical understandings, as well as them exhibiting the inquiry behaviours of: conjecturing, experimenting, creating, and communicating. Transforming games into investigations offers the potential for fostering students' mathematical proficiencies and achievement across all ages.

Problem-solving is a crucial proficiency developed through investigations into mathematical games. Setiyadi et al. [44] employed an ethnomathematics nuanced problem-based learning (PBL) model to explore traditional games, with the aim of enhancing primary students' problem-solving capabilities. Investigating areas through checkers and hopscotch is a unique approach to utilising games for investigative purposes; the games provided a hook into the mathematical concept to pique students' interest and engagement in mathematics. The PBL model experimental group demonstrated improved problem-solving capabilities over their counterparts. Games and problem-solving are effective complementary pedagogical approaches to creating a narrative-hook to excite students [10].

### 1.7. The Current Study

Although several principles of educationally rich mathematical games can be clearly identified from the literature (e.g., [9,10]), the relative value that educators place on aspects of each of these principles, as well as how these principles interact with the intended use in the classroom, has not been a focus of existing research. Undertaking such research is critical for deepening our understanding of educator decision-making in relation to using mathematical games as a pedagogical tool. Consequently, the purpose of the current study is to draw on these six principles of educationally rich mathematical games to establish which specific characteristics of a game are most important for supporting mathematics learning, from an educator's perspective. In addition, we intend to uncover those characteristics that influence an educator's intent to use the game in a classroom in the future. Our two research questions are:

1. What are the characteristics of games that educators identify as important for supporting mathematics learning?
2. What are the characteristics of games that influence the likelihood of an educator using the game in a classroom with students/children if given the opportunity?

## 2. Materials and Methods

### 2.1. Participants

One hundred and twenty-two educators completed the online questionnaire. The majority of the participants were educators from Australia (101; 83%). Other countries with respondents included: Canada (10; 8%), the United States (6; 5%), New Zealand (2; 2%) and

at an international school in Japan (1; 1%). Two participants did not specify the country in which they taught.

Most participants taught or supported mathematics instruction in a primary school setting. Specifically, two thirds of participants were current classroom teachers in a primary school or equivalent setting role (83; 68%), whilst around one sixth of participants were in a non-classroom-based mathematics leadership role in a primary school (20; 16%). The remaining participants included: pre-service teachers (4; 4%), secondary school classroom teachers (2; 2%), pre-service teacher educators (2; 2%), two participants in the tertiary sector, and educators in a variety of other roles (e.g., school principal, mathematics consultant, instructional coach, tutor). Collectively, our study participants will be referred to as educators throughout the manuscript.

### 2.2. Procedure

A questionnaire was designed through an online survey platform, Qualtrics. Convenience sampling and social media were used to disseminate the questionnaire to educators. The first author disseminated the questionnaire link after running three separate professional learning workshops targeting primary school teachers, all of which included a focus on mathematical games. Workshop participants were exposed to a broad range of mathematical games (between five and ten, depending on the specific workshop). In addition, all authors utilised social media (Twitter, Facebook) to directly connect with educators, whilst the fourth author directed participants to his YouTube channel, which contained video demonstrations of over 100 mathematical games. When completing the questionnaire, participants were invited to select any one mathematical game that they were familiar with from any relevant context and evaluate this particular game. The questionnaire was completed anonymously, and only completed questionnaires were analysed.

### 2.3. Measures

2.3.1. Measuring the Characteristics of Educationally Rich Mathematical Games (CERMaGs)

Twenty Likert items were designed to assess the extent to which an educator viewed a particular mathematical game of their choosing as educationally rich. These 20 items covered the six principles of educationally rich mathematical games outlined in the literature review. The process for designing the items involved a discussion amongst the authors as to how a particular principle of educationally rich mathematical games might be assessed. For all principles, except Principle 4: flexibility for learning and teaching, three items were designed to comprehensively assess a given principle, with one of these items being 'negatively worded'. By contrast, we deemed it necessary to design five items to appropriately assess Principle 4. The 20 items are presented in Table 1.

Prior to the 20 items being presented, the participants were first asked to nominate a mathematical game to evaluate, to describe the context in which they had encountered the game (watching an online video of the game, playing the game in a workshop, observing students playing the game in a classroom, playing the game in a home environment, other), whether they had used the game previously with children/students, and to note the year levels they thought the game was best suited for (Foundation/Kindergarten to Year 12). The participants were provided with the following prompt: With students from this year level(s) in mind, please indicate the extent to which you agree or disagree with the following statements. The 20 items from Table 1 were presented, with the participants indicating their response to each item on a 5-point scale: strongly disagree (1), disagree (2), neutral (3), agree (4), and strongly agree (5).

**Table 1.** Characteristics of educationally rich mathematical games (CERMaGs) measure.

| Principle | Items |
|---|---|
| Students are engaged | 1a The game will engage students |
| | 1b Students will enjoy playing the game |
| | 1c Students playing the game will likely get off task ˆ |
| Skill and luck | 2a The game gives all students a chance to win |
| | 2b The game represents a good balance between skill and luck |
| | 2c The game allows more skilful players to dominate ˆ |
| Mathematics is central | 3a Important mathematical ideas are central to gameplay |
| | 3b The game encourages mathematical discussion between students during gameplay |
| | 3c Mathematics seems 'tacked-on' to the game ˆ |
| Flexibility for learning and teaching | 4a The game is suitable for learners of different ability levels |
| | 4b The game can be easily modified to cater to a variety of different learners |
| | 4c The game offers a good level of challenge for students |
| | 4d Many students who struggle with mathematics would find this game too challenging ˆ |
| | 4e High performing students would find the game too easy ˆ |
| Home-school connections | 5a The game requires minimal special materials and set-up |
| | 5b The game offers a good opportunity for building connections between home and school |
| | 5c The game is hard to explain and describe to a non-teacher ˆ |
| Games into investigations | 6a The game could lead to a rich mathematical investigation |
| | 6b The game could be used to launch a mathematical inquiry |
| | 6c I think the game is more suited to a 'warm-up' or 'quick game' than a deep exploration of mathematical ideas ˆ |

ˆ negatively worded items.

### 2.3.2. Perceived Game Value

Study participants were asked a single question to determine the perceived value of their chosen game to support mathematics learning. Specifically, participants were asked: "Overall, on a scale of 1 to 10, how valuable do you think this game is for supporting students' mathematical learning? (1 = not at all valuable; 10 = extremely valuable)?". A follow-up, open-ended prompt accompanying this question was provided, "Please explain your response".

### 2.3.3. Intentions to Use the Game

Study participants were asked a single question to establish the likelihood that they would use their chosen game with students. Specifically, participants were asked: "On a scale of 1 to 10, if teaching the relevant year level, how likely would you be to use this game in your classroom? (1 = extremely unlikely; 10 = extremely likely)?". Again, the participants were provided with a follow-up, open-ended prompt, "Please explain your response".

### *2.4. Data Analysis*

SPSS v. 25 was used to examine the correlations between the characteristics of educationally rich mathematical games, perceived game value, and intentions to use the game, in order to answer our two research questions. A correlational analysis allows us to discern those game characteristics that are most strongly associated with the educator's perceptions of game value and intentions, and therefore most salient from an educator's perspective, when it comes to making decisions about the specific games to use with students. As all data was ordinal, Spearman's rank order correlation was employed (with correlations denoted by the symbol ρ).

## 3. Results

### *3.1. Descriptive Statistics*

In total, 119 educators selected 53 different games to be evaluated (three educators indicated games by the fourth author in general from his YouTube channel, rather than one specific game). Sixteen games were nominated by multiple participants, with only two

games nominated by more than seven participants (Choc-Chip Cookies 21; Get Out of My House 18).

When asked to describe the context in which they had most recently observed or played the game they were evaluating, educators indicated they had generally observed students playing the game in a classroom (49; 40%), had played the game themselves in a professional learning context (37; 30%), or had watched an online video clip of the game (28; 23%) (see Table 2).

**Table 2.** Context in which the game was most recently observed/played.

| Setting | Frequency |
|---|---|
| Observing students playing the game in a classroom | 49 (40%) |
| Playing the game in a workshop, tutorial, or other professional learning context | 37 (30%) |
| Watching an online video clip, such as YouTube, Vimeo, etc. | 28 (23%) |
| Other | 8 (7%) |

Educators were evenly split between those who had actually used the game they were evaluating with students/children themselves (62; 51%), or had not yet tried the game (60; 49%). When asked to indicate the year levels for whom the game was suitable, most participants nominated multiple year levels (114; 93%). The mean number of year levels nominated was 4.2, whilst the median number was 4. Around three-quarters of educators (76%) thought the game they were evaluating was suitable for Year 3 students (see Figure 1). This was followed by Year 4 students (71%), Year 2 students (61%) and Year 5 students (58%).

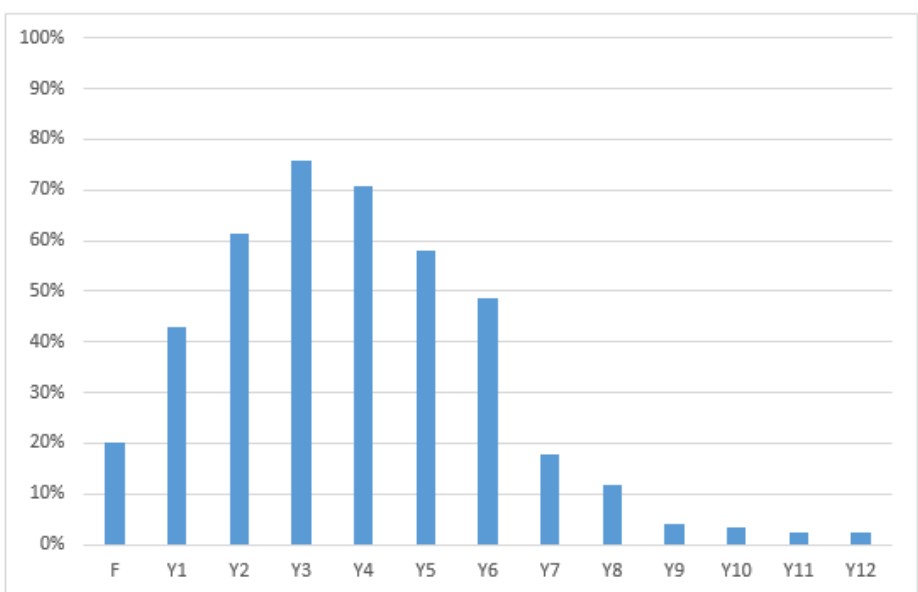

**Figure 1.** Percentage of educators indicating whether their nominated game was suitable for each year level.

Overall, as revealed in Table 3, educators perceived their chosen game as highly valuable for supporting mathematics learning and were highly likely to use the game in a classroom in the future if afforded the opportunity. Importantly, the scores on our scales of perceived game value and intentions to use game were both very high, independent of whether participants had used the game in a classroom with students.

**Table 3.** Perceived game value and intentions to use game.

|  | Mean (Out of 10) | SD | Median |
|---|---|---|---|
| Perceived game value (all educators) | 8.95 | 1.21 | 9 |
| Perceived game value (used game already) | 8.97 | 1.09 | 9 |
| Perceived game value (not yet used game) | 8.93 | 1.34 | 9 |
| Intentions to use the game (all educators) | 9.57 | 1.04 | 10 |
| Intentions to use (used the game already) | 9.73 | 0.93 | 10 |
| Intentions to use (not yet used the game) | 9.42 | 1.12 | 10 |

### 3.2. Describing the Characteristics of the Educationally Rich Mathematical Games (CERMaGs) Measure

Although it is not the primary purpose of the current paper to use the characteristics of educationally rich mathematical games (CERMaGs) items as a single measure, nor to establish its psychometric properties, we report this information here as it may be of use for future research in this space. However, we note that further work needs to be carried out before we can conclude that the CERMaGs items operate as a valid and reliable measure (e.g., factor analysis, incorporating a larger sample of participants).

With 20 items, each measured on a 5-point scale (1 = strongly disagree, 2 = disagree, 3 = neutral, 4 = agree, 5 = strongly agree), the maximum possible score on the CERMaGs measure is 100 and the minimum score is 20. The reliability coefficient (Cronbach's alpha) was good ($\alpha = 0.80$). An exploration of the scale revealed one outlier value (CERMaGs = 54). With this outlier removed, the CERMaGs measure was approximately normally distributed. The mean score on the CERMaGs measure was 83.3, the median score was 83, and the standard deviation was 6.9. The minimum actual score on the CERMaGs measure was 63, whilst the maximum actual score was 97.

As an indicator of its construct validity, we expected the CERMaGs measure to be correlated positively with the educators' perceptions of a game's capacity to support mathematics learning, as well as the likelihood that they would use the game with students/children in a classroom. As noted in the methodology section, this information was collected from participants using single-item measures, neither of which was normally distributed. Indeed, further analysis revealed that the CERMaGs measure shared a medium positive correlation both with the educators' perceptions of a game's mathematical learning capacity ($\rho = 0.41$, $p < 0.05$) and the likelihood that an educator would use the game in their classroom, if given the opportunity ($\rho = 0.32$, $p < 0.05$).

One further point to note regarding CERMaGs as a measure. We might expect educators who have used the game they evaluated with students/children to report the game as more educationally rich than those educators who had not. This might either be because educators have a stronger sense of the game's affordances having used it in practice, or because a previous positive evaluation of the game led to them deciding to use it in practice prior to completing the questionnaire. We do in fact find that those educators who report that they have used the game they are evaluating with students/children (Mean = 85.0, SD = 6.2), evaluate the game more positively on the CERMaGs measure than those who have not (Mean = 81.6, SD = 7.2), $t(119) = 2.76$, $p < 0.01$. The effect size for this analysis ($d = 0.51$) was moderate, following Cohen's [45] convention.

Finally, we examined whether the CERMaGs measure was related to the number of year levels that educators nominated the game as being suitable. Again, it seems probable that games that can be used with a greater variety of year levels will be perceived as more educationally rich than games perceived as having a narrower year level focus. This is indeed what we found. Specifically, there was a medium positive correlation between the CERMaGs measure and the number of year levels that educators indicated the game could potentially be used with ($\rho = 0.34$, $p < 0.05$).

### 3.3. Characteristics of Games That Educators Identify as Important for Supporting Mathematics Learning

In order to answer our first research question, we investigated the relationship between the various characteristics of educationally rich games, operationalised as the 20 items presented in Table 1, and educators' perceptions of the extent to which their chosen game supported mathematics learning. Table 4 presents the correlations between the various game characteristics and educators' perceptions of a game's value for supporting learning. In addition, Table 5 presents those characteristics of games that shared at least medium sized correlations ($\rho > 0.30$; see [46]) with perceptions of learning, as well as exemplar quotations representative of these characteristics taken from the qualitative data inviting participants to explain their rating regarding the value of their chosen mathematical game for supporting learning.

**Table 4.** Correlations between game characteristics, game value, and the intentions to use the game.

| Principle | Items | Game Value ($\rho$) | Intentions to Use ($\rho$) |
|---|---|---|---|
| Students are engaged | 1a The game will engage students | 0.36 * | 0.30 * |
| | 1b Students will enjoy playing the game | 0.34 * | 0.34 * |
| | 1c Students playing the game will likely get off task ^ | −0.15 | −0.15 |
| Skill and luck | 2a The game gives all students a chance to win | 0.19 * | 0.28 * |
| | 2b The game represents a good balance between skill and luck | 0.10 | 0.04 |
| | 2c The game allows more skilful players to dominate ^ | −0.01 | −0.17 |
| Mathematics is central | 3a Important mathematical ideas are central to gameplay | 0.21 * | 0.15 |
| | 3b The game encourages mathematical discussion between students during gameplay | 0.28 * | 0.30 * |
| | 3c Mathematics seems 'tacked-on' to the game ^ | −0.19 * | −0.15 |
| Flexibility for learning and teaching | 4a The game is suitable for learners of different ability levels | 0.19 * | 0.24 * |
| | 4b The game can be easily modified to cater to a variety of different learners | 0.39 * | 0.28 * |
| | 4c The game offers a good level of challenge for students | 0.39 * | 0.17 |
| | 4d Many students who struggle with mathematics would find this game too challenging ^ | −0.10 | −0.18 * |
| | 4e High performing students would find the game too easy ^ | −0.22 * | −0.14 |
| Home-school connections | 5a The game requires minimal special materials and set-up | 0.18 * | 0.13 |
| | 5b The game offers a good opportunity for building connections between home and school | 0.17 | 0.13 |
| | 5c The game is hard to explain and describe to a non-teacher ^ | −0.02 | −0.14 |
| Games into investigations | 6a The game could lead to a rich mathematical investigation | 0.36 * | 0.30 * |
| | 6b The game could be used to launch a mathematical inquiry | 0.30 * | 0.15 |
| | 6c I think the game is more suited to a 'warm-up' or 'quick game' than a deep exploration of mathematical ideas ^ | −0.11 | 0.08 |

^ negatively worded items. * $p < 0.05$.

**Table 5.** Games value for supporting learning: exemplary quotations for characteristics with at least medium-sized correlations.

| Characteristic | Exemplary Quotation |
|---|---|
| Principle 4c: The game offers a good level of challenge for students | "The kids will be able to be challenged and (we) will be able to extend and enable students." |
| Principle 4b: The game can be easily modified to cater to a variety of different learners | "So many ways to adapt the game to suit learners." |
| Principle 6a: The game could lead to a rich mathematical investigation | "I think there are a lot of strong underlying mathematics ideas that can be explored in this game. It is also adaptable and allows for deeper investigation." |
| Principle 1a: The game will engage students | "The level of engagement with this game is astounding. I work with very reluctant students and it was this game that 'won' them over!" |
| Principle 1b: Students will enjoy playing the game | "Students showed a high level of enjoyment (playing the game)" |
| Principle 6b: The game could be used to launch a mathematical inquiry | "The game could also be used as a springboard for inquiry into other areas of maths, such as capacity and volume, given the right questions and materials." |

The most important characteristics for determining a game's value related to whether the game offered a good level of challenge for students, and whether it could be easily modified to cater to different learners. In addition, whether students were engaged in the game and appeared to enjoy playing it were closely associated with perceptions of a game's value, as was the potential for a game to lead to an investigation or to launch a rich mathematical inquiry. The other notable characteristic that approached a medium-sized correlation with a game's overall value was whether the game generated a mathematical discussion between students whilst playing. By contrast, whether a game was too hard for some learners, or unfair in the sense that more skilful players might dominate, were not important considerations when determining a game's potential value for mathematics learning.

*3.4. Characteristics of Games That Influence the Likelihood of an Educator Using the Game*

The extent to which the various characteristics of educationally rich games were related to the likelihood that an educator would use the game in a classroom were examined to answer the second research question. Table 4 presents the correlations between game characteristics and the likelihood of an educator using a game. In addition, Table 6 presents those characteristics of games that shared at least medium-sized correlations ($\rho > 0.30$; see [46]) with likelihood of use, as well as exemplar quotations representative of these characteristics taken from the qualitative data inviting participants to explain their rating of how likely they would be to use the game in a classroom with students/children, if given the opportunity.

**Table 6.** Intentions to use game: exemplary quotations for characteristics with at least medium-sized correlations.

| Characteristic | Exemplary Quotation |
|---|---|
| Principle 1b: Students will enjoy playing the game | "I have used it many times! My students love it!" |
| Principle 3b: The game encourages mathematical discussion between students during gameplay | "It would allow for rich discussion amongst students and I liked how playing against the teacher allows students to collaborate and share ideas" |
| Principle 6a: The game could lead to a rich mathematical investigation | "It would provide (an) engaging activity when working on multiplication as well as opportunities for investigation." |
| Principle 1a: The game will engage students | "There is maximum engagement for student and opportunities for students to explain their thinking and strategies" |

In a similar manner to the characteristics closely associated with a game's value for supporting learning, whether a game was perceived as enjoyable to play and engaging were important factors in determining whether an educator intended to use the game with students in the future. In addition, the extent to which a game generated mathematical discussion was again important. Other characteristics that were approaching a medium-sized correlation included the extent to which the game could be modified for different learners, and whether the game gave all students a chance to win. Interestingly, none of the three items representing Principle 5, 'home-school connections', were associated with the likelihood that an educator would use a game. This finding suggests that factors such as accessing materials and setting up the game were not important considerations for teachers in determining the potential usability, at least in relation to the game they chose to evaluate.

## 4. Discussion and Conclusions

The current study drew upon the six principles of educationally rich mathematical games previously identified in the literature [9,10], to explore how the specific characteristics of mathematical games related to their value for supporting learning and the intentions for educators to use a game with their students in the future. The extent to which educators perceived a game to be suitably challenging, engaging, enjoyable, modifiable to support different learners, and transformable into an investigation or broader mathematical inquiry,

were particularly important characteristics associated with perceptions of educational value. Similarly, perceived levels of student enjoyment, engagement, and a game's potential to lead to a rich mathematical investigation, were important characteristics for evaluating the likelihood that an educator would use a particular game in the future with students if given the opportunity, as was the capacity of a game to support mathematical discussion.

A limitation of the current study was the use of a convenience sample, which meant that our study participants were not representative of a broader population of educators. An additional, less obvious but equally important, limitation was that, as educators were invited to choose any game to evaluate for the purposes of completing the questionnaire, they tended to choose games they perceived as highly valuable and that they intended to use in the classroom. In a sense, this was a strength of the study, as it could be argued that it is important to study educator's reactions to what are considered excellent games, as opposed to mathematical games "out in the wild", as such games are more reflective of the types of activities educators should be encouraged to incorporate into their practise. However, it also served to limit the level of variability in the responses, which somewhat undermined our efforts to establish those characteristics of games most strongly associated with game utility. To increase response variability, a future study may instead present a series of games to a group of educators, some of which appear to be objectively 'better' than others, and invite participants to complete the questionnaire on multiple occasions to enable them to compare and contrast each of these games. Finally, although we think that the relative broad range of games included in our study (53 different games) allows us to draw some robust conclusions about the 'average' correlations between game characteristics, game value, and intentions to use, our study design is not sensitive to the possibility that the size and direction of these correlations might vary across different types of games. Again, the exploration of interaction effects between specific games (or game types) and the strength and direction of these correlations would require a different research design (e.g., a small number of games or game-types being systematically compared across a large number of participants). This is another possibility for a future study.

The paper concludes by discussing some of the potential practical implications arising from our study, which were highlighted by the third and fourth authors, who both work actively in primary schools. These implications include: the value of educationally rich mathematical games across multiple year levels, understanding how games support mathematics discussion, games are 'more than fun' and professional learning to support home-school connections.

### 4.1. Value of Educationally Rich Mathematical Games across Multiple Year Levels

Overall, educators in this study have identified games with a high perceived value for use in the classroom, and the vast majority indicated that these games are suitable for use with multiple-year levels, with a mean of 4.2 year levels for each game nominated. The value in adapting mathematical games for students across age groups and readiness levels appears well understood by mathematical educators and leaders participating in this study, an idea that is not always reflected in practice within schools. Specifically, some schools can work in relative year-level 'silos' [47], whereby teachers lack an understanding of how games might be adapted across multiple year-level teams, and there is limited sharing of resources between these teams. There can be the perception that certain mathematical concepts or gameplay can be overly complex for young students or conversely, games can be too simple for older students. The value of an educationally rich game that can be adapted to engage students across many year levels is an important point of emphasis for mathematics leaders and consultants, as it promotes vertical collaboration within a school, encourages the sharing of quality resources and, through games that can be highly differentiated, focuses practice on educational readiness rather than age groups. There may be an opportunity to further explore the features of games that fall into this category, compared with those games that are perceived to be narrower in terms of their scope.

This finding has implications for the way in which primary school mathematical games might be utilised as a part of a professional learning program for secondary schools. There is scope to better understand how games are currently used to engage secondary school learners, and how the resources and pedagogical approaches at a primary level may be used to augment current practice.

### 4.2. Understanding How Games Support Mathematical Discussion

The extent to which a game promotes mathematical discussion was identified as an important factor in determining the likelihood that an educator would use a game in the classroom. Games have a unique role in the classroom to promote purposeful dialogue around mathematical concepts, with players often required to articulate and justify their mathematical reasoning. Despite educators indicating that discussion is an important feature of mathematical games, there is currently limited research into the level or nature of this mathematical discussion, or how games compare with other pedagogical approaches, in terms of promoting purposeful dialogue [48]. There is an opportunity to further explore the features of those games that promote rich mathematical discussion, including whether the game is played individually or as a partnership/team and whether the game is adversarial (student/s against student/s) or not (student/s with a specific objective/playing against 'the game').

### 4.3. Games Are 'More Than Fun'

Although student engagement and enjoyment are key factors in educators choosing mathematical games, this study confirms the earlier research that the reasons for game selection are much broader [4]. In this study, a game's capacity to be differentiated and to challenge all learners were also identified as important characteristics of educationally rich games. Additionally, educators in this study highlighted their chosen games' value as a prompt for a deeper investigation or to launch a mathematical inquiry.

Despite the widespread use of games in primary schools throughout Australia, there is still a perception amongst some teachers and leaders within the school system that the core value of games is enjoyment. The convenience sample used for this study does not capture the extent of this perception within schools across the country. It is important that through professional learning and teacher-preparation programs, that the broad benefits of mathematical games are understood, and that educators are provided with the skills to choose and adapt games to challenge their learners. Many teachers lack access to resources or the confidence to explore how to utilise a game for deeper investigations or mathematical inquiry, and this presents a further professional learning opportunity.

### 4.4. Supporting Home-School Connections

Finally, this research has corroborated earlier work suggesting that educators place less emphasis on the value of games for building home-school connections than other principles of educationally rich mathematical games [4]. In the current study, this association manifested as a non-significant correlation between whether the game provided opportunities for building connections between home and school, and both the educational value of a game, and future intentions to use a particular game. It might be useful for teachers to participate in further professional learning supporting them to consider the value of a game beyond the classroom, in a similar manner to how encouraging and resourcing home reading is viewed as a significant component of developing literacy [49].

**Author Contributions:** Conceptualization, all authors.; methodology, J.R. and L.B.; formal analysis, J.R.; writing—original draft preparation, L.B. and J.R.; writing—implications, T.R. and M.M.; writing—revising manuscript, all authors. All authors have read and agreed to the published version of the manuscript.

**Funding:** This research received no external funding.

**Institutional Review Board Statement:** The study was conducted according to the guidelines of the Declaration of Helsinki, and was approved by the Ethics Committee of Monash University (Project 21806).

**Informed Consent Statement:** Informed consent was obtained from all subjects involved in the study.

**Data Availability Statement:** Data is available from the authors on request.

**Conflicts of Interest:** The authors declare no conflict of interest.

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
