# Peer review of "Identifying the Characteristics of Non-Digital Mathematical Games Most Valued by Educators"

_education, doi:10.3390/educsci13010030_

Round 1

Reviewer 1 Report

Thank you for your submission and sharing your research study. Here are some suggestions for improving the quality of the article:

1) Games are often seen in the US and other countries as "fluency games" in mathematics education. Work by Bay-Williams, Kling, and others should be reviewed and incorporated into the literature review.

2) Further, there is a need for a clear and coherent argument about why teachers' preference or choice in games is helpful. For example, are games used only if teachers like them? 
Are they not given or required to use specific resources or do they have complete autonomy? 

3) Based on the literature review what is the gap and area in the literature that this study is attempting to cover?

4) It seems peculiar that the 4th author did workshops on games and then this survey was done- was there no bias or conflict of interest? Was this an unbiased or biased sample? What influence does this sample have on the findings? Are the findings reflective of teachers in general? 

5) The methods- please explain the rationale for using correlations. What is it about your research questions that led to this.

6) Findings- statistically significant correlations mean what exactly.. make sure you interpret and help the reader! 

7) Discussion- how does this study and the findings contribute to the field? what is the impact of this study? 

8) what are future studies that are needed now? the link between games and learning still is under-studied and needs further examination! 

Author Response

Reviewer 1

Response: Thank you for your review. Please see our response to your comments below.

Thank you for your submission and sharing your research study. Here are some suggestions for improving the quality of the article:

1) Games are often seen in the US and other countries as "fluency games" in mathematics education. Work by Bay-Williams, Kling, and others should be reviewed and incorporated into the literature review.

Response: The work of Bay-Williams and Kling had been included in the literature review, and further work by these authors has been incorporated. Several other additional references have been included to further bolster the literature review.

2) Further, there is a need for a clear and coherent argument about why teachers' preference or choice in games is helpful. For example, are games used only if teachers like them? 
Are they not given or required to use specific resources or do they have complete autonomy? 

Response: Teachers in Australia have the autonomy to select the resources they employ in the classroom, and the program they develop. Australian teachers’ preferences impact their pedagogical and content choices.This point has now been incorporated into the introduction, with citations to support the argument.

3) Based on the literature review what is the gap and area in the literature that this study is attempting to cover?

Response: We have expanded the section under “The Current Study” to make it clearer the gap in the literature that our study is endeavouring to address. The key point is that although several principles of educationally-rich mathematical games can be clearly identified from the literature, the relative value that educators place on aspects of each of these principles, as well as how these principles interact with intended use in the classroom, has not been a focus of existing research.

4) It seems peculiar that the 4th author did workshops on games and then this survey was done- was there no bias or conflict of interest? Was this an unbiased or biased sample? What influence does this sample have on the findings? Are the findings reflective of teachers in general?

Response: As noted in the method section, the questionnaire was completed online and anonymously, therefore limiting any social desirability bias etc. associated with some responses being in a workshop setting. However, as noted in the Discussion and Conclusions, the use of a convenience sample does indeed limit the generalisability of our study, whilst the fact that the games were very highly rated on average (potentially in part an artefact of some participants completing the questionnaire after attending a PL workshop) limits response variability (which somewhat undermined our efforts to establish those characteristics of games most strongly associated with game utility).

5) The methods- please explain the rationale for using correlations. What is it about your research questions that led to this.

Response: We have now noted in the methods section 2.4 the benefits of using correlational analysis for addressing our study aims/ research questions.

6) Findings- statistically significant correlations mean what exactly.. make sure you interpret and help the reader! 

Response: Rather than only refer to whether or not a correlation is statistically significant, we have emphasised effect size (e.g., a medium-size correlation) throughout the results section, as we believe this to be of more practical significance (given that the statistical significance of a correlation does not mean it is of practical importance, and that statistical significance is obviously sensitive to sample size).

7) Discussion- how does this study and the findings contribute to the field? what is the impact of this study? 

Response: Sections 4.1-4.4 elaborate on some of the implications of our study, what it means for classroom practice and implications for future research.

8) what are future studies that are needed now? the link between games and learning still is under-studied and needs further examination! 

Response: We have noted that games remains an under-researched area, and have posed specific suggestions for future research that could be undertaken to further unpack the findings from our research. For example: “Despite educators indicating that discussion is an important feature of mathematical games, there is currently limited research into the level or nature of this mathematical discussion, or how games compare with other pedagogical approaches in terms of promoting purposeful dialogue (Heshmati et al., 2018). There is an opportunity to further explore the features of those games that promote rich mathematical discussion, including whether the game is played individually or as a partnership/team and whether the game is adversarial (student/s against student/s) or not (student/s with a specific objective / playing against ‘the game’).”.

Reviewer 2 Report

The manuscript is well written. It is very clear and well organized. The study is convincing with clear methodology and significant findings. However, there are some clarifications needed to be addressed before publication.

(1) Focus of the study - The authors mentioned that “… which characteristics of educationally rich mathematical games are most valued by educators.” I think it would be good if the authors could explain further why this is important?  For instance, would this lead to more development of digital or non-digital games in the future? From a reader’s point of view, I could see the significance of the study.

(2) CERMaG  - I am concern with the “perceived game value and intentions to use game”. For example, educators were asked about “Students are engaged”. Some educators have yet used games, so how did they know the students will enjoy playing the games? Another example - “Skill and luck”. They would not know “The game gives all students a chance to win” if they never tested it on their students.

(3) Game – In this study, “… participants were first asked to nominate a mathematical game to evaluate”.  Although the authors have addressed this as one of the limitations, I think different games may lead to different outcomes (e.g., correlations). For example, game A (Mathematics is central) and perceived game value might be positively correlated but game B (Mathematics is central) and perceived game value might be negatively correlated. How the authors address this inconsistency?

(4) Table 4 – In this table, the authors presented the “Correlations between Game Characteristics, Game Value and Intentions to Use Game”. The correlation analysis was performed on each item (one single question but not according to the principle). Why was it performed this way?

This manuscript is well presented, and I really enjoy reading it.

Author Response

Reviewer 2

Response: Thank you for your review. Please see our response to your comments below.

The manuscript is well written. It is very clear and well organized. The study is convincing with clear methodology and significant findings.

Response: Thank you for the positive feedback.

However, there are some clarifications needed to be addressed before publication.

(1) Focus of the study - The authors mentioned that “… which characteristics of educationally rich mathematical games are most valued by educators.” I think it would be good if the authors could explain further why this is important?  For instance, would this lead to more development of digital or non-digital games in the future? From a reader’s point of view, I could see the significance of the study.

Response: We have added a couple of points to emphasise why the study is significant. First, teachers’ valuing of, and preferences for utilising, specific types of mathematical learning activities is a particularly relevant consideration in countries such as Australia, where the curriculum is not prescriptive and teachers have the autonomy to develop a specific learning program that meets their students’ needs. Second, although several principles of educationally-rich mathematical games can be clearly identified from the literature, the relative value that educators place on aspects of each of these principles, as well as how these principles interact with intended use in the classroom, has not been a focus of existing re-search. Undertaking such research is critical for deepening our understanding of educator decision-making in relation to using mathematical games as a pedagogical tool.

(2) CERMaG  - I am concern with the “perceived game value and intentions to use game”. For example, educators were asked about “Students are engaged”. Some educators have yet used games, so how did they know the students will enjoy playing the games? Another example - “Skill and luck”. They would not know “The game gives all students a chance to win” if they never tested it on their students.

Response: The point that only around half educators (51%) have used these games with students is a valid concern. However, effectively all participants responding to the questionnaire had seen the game played or played the game themselves in one context or another. Moreover, the focus of our research is on understanding how participants perceived these various characteristics to be related to game value and intentions to use in the classroom; that is, their internal mental model about what makes an educationally-rich mathematical game (e.g., it has to be engaging; it has to offer a good balance between skill and luck). In this sense, the specific game being evaluated is less important than obtaining an overall sense about how educators perceive these various constructs to be interrelated.

(3) Game – In this study, “… participants were first asked to nominate a mathematical game to evaluate”.  Although the authors have addressed this as one of the limitations, I think different games may lead to different outcomes (e.g., correlations). For example, game A (Mathematics is central) and perceived game value might be positively correlated but game B (Mathematics is central) and perceived game value might be negatively correlated. How the authors address this inconsistency?

Response: Thanks for this point. As you noted, we have addressed this in our limitation section and we agree that further research with a more diverse variety of games is worthwhile. To the extent that it is possible that the size and direction of these correlations might vary across different types of games, we think that the relative broad range of games included (53 different games) allows us to draw some robust conclusions about ‘average’ correlations across a broad variety of games. The issue you raise - essentially involving an exploration of the interaction effects between specific games and the strength/ direction of these correlations - would obviously require a different research design (e.g., 3 games being compared across 120 participants). This is another possibility for a future study. We have added this additional point to our limitations section.

(4) Table 4 – In this table, the authors presented the “Correlations between Game Characteristics, Game Value and Intentions to Use Game”. The correlation analysis was performed on each item (one single question but not according to the principle). Why was it performed this way?

Response: When considering the literature, it is clear that each of these principles has multiple related elements (which we have articulated as ‘characteristics of educationally rich mathematical games). We considered aggregating each of the characteristics to provide a total for each principle, however concluded that such a process would likely suggest the necessity for factor analysis or something similar (i.e., do these 20 characteristics actually load on the 6 principles). Given the sample size and number of characteristics would not necessarily support this sort of analysis, and the fact that our research is novel and exploratory, we decided to keep the analysis at the item level. This is one of the reasons that we present the CERMaG as a measure only tentatively (because we have not done this more rigorous work). In response to this comment by the reviewer, we have added in an additional sentence in Section 3.2, noting that further work needs to be done before we can conclude that the CERMaG is a valid and reliable measure.

This manuscript is well presented, and I really enjoy reading it.

Response: Thank you for the positive feedback.

Reviewer 3 Report

The article presents a well-researched and interesting look at what sorts of characteristics of games are valued by educators. The presented characteristics of educationally-rich mathematical games are well-researched and in line with game design and educational principles.

In both the abstract and before line 211 (or within that paragraph), it needs to be made clearer that the games described by participants were already excellent games, so that a follow up study on a larger variety of games is essential before broad conclusions can be made. This is correctly noted in the conclusion, but also needs to be up front in the abstract.

Within the paragraph of line 211, the ways in which participants chose a game to describe should be made clearer. The fact that they were in a workshop with one of the authors first and what games they could choose from, needs to be made explicit.

Finally, it would also be interesting to hear from the authors why it is important to study educator reactions to these excellent games, as opposed to math games “out in the wild.” Something about how exposing them to how good a math game can be a way to see what they really value?

Very minor edits:

On line 37 it should be noted that cost and access to technology likely also play a part in preference for non-digital games.

Educationally-rich is inconsistently hyphenated. It should be hyphenated when in front of the word games, but otherwise not (so, for example, it should not be hyphenated on line 5, but should be hyphenated on line 200).

In table 1, the use of a ^ for a footnote is odd. Usually a * is used.

Having each part of that table centered made it difficult to read. I know, that's in their template, but maybe they would let you change it anyway? They do say that the format is flexible.

Author Response

Reviewer 3

Response: Thank you for your review. Please see our response to your comments below.

The article presents a well-researched and interesting look at what sorts of characteristics of games are valued by educators. The presented characteristics of educationally-rich mathematical games are well-researched and in line with game design and educational principles.

Response: Thank you for the positive feedback.

In both the abstract and before line 211 (or within that paragraph), it needs to be made clearer that the games described by participants were already excellent games, so that a follow up study on a larger variety of games is essential before broad conclusions can be made. This is correctly noted in the conclusion, but also needs to be up front in the abstract.

Response: We have added this point to the abstract in response to this comment.

Within the paragraph of line 211, the ways in which participants chose a game to describe should be made clearer. The fact that they were in a workshop with one of the authors first and what games they could choose from, needs to be made explicit.

Response: This discussion has been extended to make it clearer that participants were free to choose any game they liked to evaluate, and to emphasise that these workshops were exposing participants to a broad rather than a narrow range of games.

Finally, it would also be interesting to hear from the authors why it is important to study educator reactions to these excellent games, as opposed to math games “out in the wild.” Something about how exposing them to how good a math game can be a way to see what they really value?

Response: This is an interesting point, and not something that we had considered when we designed the study (because we were not intending for the games to be perceived as excellent per se). We have now noted this as a point in our discussion.

Very minor edits:

On line 37 it should be noted that cost and access to technology likely also play a part in preference for non-digital games.

Response: This point has been added.

Educationally-rich is inconsistently hyphenated. It should be hyphenated when in front of the word games, but otherwise not (so, for example, it should not be hyphenated on line 5, but should be hyphenated on line 200).

Response: Thanks for pointing this out. We have now made it consistent in line with this suggestion.

In table 1, the use of a ^ for a footnote is odd. Usually a * is used.

Response: We have used ^ to distinguish a more general note from *, used later on to describe when a correlation was statistically significant.

Having each part of that table centered made it difficult to read. I know, that's in their template, but maybe they would let you change it anyway? They do say that the format is flexible.

Response: We will raise this with the editor.